# Predictive Accuracy of Singleton Versus Customized Twin Growth Chart for Adverse Perinatal Outcome: A Cohort Study

**DOI:** 10.3390/ijerph18042016

**Published:** 2021-02-19

**Authors:** Urszula Nowacka, Katarzyna Kosińska-Kaczyńska, Paweł Krajewski, Aleksandra Saletra-Bielińska, Izabela Walasik, Iwona Szymusik

**Affiliations:** 1Department of Obstetrics and Gynecology, Medical University of Warsaw, Starynkiewicza Square 1/3, 02-015 Warsaw, Poland; ulasarzynska@gmail.com (U.N.); pawelkrajewski29@gmail.com (P.K.); aleksandra.saletra@gmail.com (A.S.-B.); iwona.szymusik@wum.edu.pl (I.S.); 2Department of Obstetrics and Gynecology, Institute of Mother and Child, Kasprzaka 17a, 01-211 Warsaw, Poland; 3Department of Obstetrics and Gynecology, The Center of Postgraduate Medical Education, Cegłowska St. 80, 01-809 Warsaw, Poland; 4Students Scientific Association, Department of Obstetrics and Gynecology, Medical University of Warsaw, Starynkiewicza Square 1/3, 02-015 Warsaw, Poland; izabela.a.walasik@gmail.com

**Keywords:** twin gestation, growth chart, small for gestational age, fetal growth restriction, fetal growth

## Abstract

Background: Fetal growth of twins differs from singletons. The objective was to assess the fetal growth in twin gestations in relation to singleton charts and customized twin charts, respectively, followed by a comparison of the frequency of neonatal complications in small-for-gestational-age (SGA) twins. Methods: We performed an analysis of twin pregnancies with established chorionicity with particular emphasis on postnatal adverse outcomes in newborns classified as SGA. Neonatal birth weight was comparatively assessed using both singleton and twin growth charts with following percentile estimation. Using a statistical model, we established the prediction strength of neonatal complications in SGA twins for both methods. Results: The dataset included 322 twin pairs (247 cases of dichorionic and 75 cases of monochorionic diamniotic gestations). Utilization of twin-specific normograms was less likely to label twins as SGA—nevertheless, this diagnosis strongly correlated with risk of observing adverse outcomes. Using a chart dedicated for twin pregnancies predicted newborn complications in the SGA group with higher sensitivity and had better positive predictive value regarding postnatal morbidity. Conclusions: Estimating twin growth with customized charts provides better prognosis of undesirable neonatal events in the SGA group comparing to singleton nomograms and consequently might determine neonatal intensive care prenatal approach.

## 1. Introduction

Small-for-gestational age (SGA) describes a fetus/neonate with an estimated/actual birth weight less than the 10th centile for the corresponding gestational age. Fetal growth restriction (FGR) is an implication of a pathological retardation of genetic growth potential, which may result in fetal compromise (abnormal Doppler studies, reduced amniotic fluid volume); therefore, FGR is not synonymous with SGA [1,2]. In singleton pregnancies, implementation of centiles customized for maternal characteristics, gestational age, and infant gender identifies neonates at high risk of morbidity and mortality better than populational centiles [3].

Selecting SGA fetuses in singleton pregnancies has a major influence on the level of provided perinatal care. In comparison to appropriate for gestational age (AGA) newborns, SGA babies are at increased risk of morbidity (OR 2.26, 95% CI 1.04–4.39) [4] and unfavorable neurodevelopmental outcome at the age of 24 months (lower centiles in problem solving, 42.8 vs. 52.1 centile, *p* = 0.001, and in personal-social areas, (44.4 vs. 54.6 centile, *p* < 0.001, compared to controls) [5].

Recently, an incremental surge in twin gestation occurrence resulted in a vivid interest in the pathophysiology of multiples’ growth. Management of twin pregnancies is widely discussed due to different pathophysiology and an increased risk of stillbirth when reaching term. Avoidance of unnecessary preterm birth is particularly important in a group that is already at high risk of preterm delivery, both spontaneous and iatrogenic. Every intervention carries a risk of not only prematurity-related complications, but also brings a financial burden. Whether still unknown if it is an adaptive or physiological mechanism, a retardation of twins’ growth is usually observed around 30–32 gestational weeks, and therefore using singleton growth charts might not be adequate [6,7]. Monochorionic placentation appears to be an individual pathological factor, as monochorionic twins have lower adjusted birthweights than dichorionic when compared to singleton [8]. Moreover, it has been established that male fetuses have higher growth rate than female ones [9]. Authors of publications on twin growth use either singleton or multiple growth curves, as evidence as to which chart estimates the pace of twin growth more precisely is lacking [10,11]. The Working Group on Fetal Biometric Charts developed customized nomograms recently published in American Journal of Obstetricians and Gynaecologists [12]. They were based on measurements of fetal biometry obtained from serial ultrasound examinations in uncomplicated twin pregnancies, using multilevel linear regression models. Parental characteristics, parity, fetal sex, and mode of conception also had an impact on calculations, later adjusted to chorionicity and compared with growth curves for singletons. However, the accuracy of either singleton or twin growth charts in establishing the risk of adverse neonatal outcomes in twins is controversial. The objective of our study was to determine whether singleton or twin growth charts predict neonatal adverse outcomes in the SGA group more precisely.

## 2. Materials and Methods

This was a retrospective cohort study conducted in a tertiary reference center. Medical details and pregnancy outcomes of 489 women in twin gestation who had given birth between 2005 and 2015 in the 1st Department of Obstetrics and Gynecology of Medical University of Warsaw were analyzed. On admission, all women gave informed consent on anonymous disclosure of their medical data for further analysis. The Medical University of Warsaw Bioethics Committee approved the study protocol and the use of medical data of the admitted patients (AKBE/23/2016). The inclusion criteria contained: delivery beyond 24 + 0 weeks of gestation, chorionicity established and documented on the 1st trimester sonographic scan (two gestational sacs or lambda sign for dichorionic pregnancy; single gestational sac or T sign for monochorionic pregnancy), verified gestational age (GA), newborns’ birth weights (BW), and complete medical data on the pregnancy outcome and neonatal outcome. Pregnancies complicated by one or two fetal demises, genetic or major anatomical abnormalities, twin to twin transfusion syndrome (TTTS), twin anemia-polycythemia sequence (TAPS), twin reversed arterial perfusion syndrome (TRAP), as well as monochorionic monoamniotic pregnancies were excluded from the study. Prenatally, growth charts should refer to ideal circumstances, under which fetuses reach their maximum growth potential. In all of the above-mentioned situations, growth retardation may play a major pathophysiological role, and therefore those cases were excluded from analysis. In a case of fetal demise in a twin gestation, the exact time of the event is difficult to establish and a birth weight of a demised fetus may be falsely decreased due to the delay between the time of demise and delivery. It is rather difficult to estimate the percentage of FGR in demised fetuses in a retrospective analysis and, therefore, all the cases of in utero death were excluded from analysis. An arbitrary gestational age criterion—deliveries from 24 weeks onward, commonly used in reviewed articles—was set due to the viability threshold in neonatal care in the majority of countries worldwide. Moreover, in cases of miscarriage before 24 weeks, the patients were not routinely hospitalized, only if induction of labor was necessary. Gestational age was calculated on the basis of the first day of last menstrual period or the transfer day in assisted reproduction techniques procedures and verified by the crown-rump length (CRL) measured on the first trimester scan (if estimated due dates were inconsistent and the difference was bigger than 5 days, the ultrasound measurement was of primary importance; in case of the CRL discordance, the measurement from the larger twin was chosen). SGA was defined as BW < 10th percentile, BW between 10th and 90th was classified as AGA, and BW > 90 percentile was labelled as large for gestational age (LGA). The customized charts for twins used in our study were developed by The Working Group on Fetal Biometric Charts with requirement of at least 2 sets of longitudinal measurements for each pregnancy, including the following: biparietal diameter (BPD), fetal head circumference (HC), abdominal circumference (AC), and femur length (FL) [9]. The estimated fetal weight (EFW) was calculated using Hadlock III formula, which incorporates the HC, AC and FL (Log10 weight = 1.326 − 0.00326 AC × FL + 0.0107 HC + 0.0438 AC + 0.158 FL) [13]. A linear mixed model was used for a delineation of curve trajectories, calculating other variables like maternal and paternal weight and height, ethnic group, parity, and sex [12]. In the set analyzed by The Working Group on Fetal Biometric Charts study, more than 90% of population was Caucasian, which remained consistent with our study group. Regarding singletons, we opted for charts developed by the same authors, and therefore the reference limits used were designed in concordance with a similar data validation method [14]. Those biometric charts were customized for parental characteristics, race, and parity, using quantile regression analysis.

Adverse neonatal outcome was defined as one or more of the following: 5-min Apgar score < 8, need for intubation, need for continuous positive airway pressure or mechanical ventilation usage, Neonatal Intensive Care Unit admission, intraventricular hemorrhage grade III or IV, necrotizing enterocolitis, neonatal pneumonia or inborn infection, or neonatal death during first 28 days of life. For each outcome, a corresponding group of affected neonates was assigned, with regard to the birth weight and a normogram used.

Statistical analysis was performed with the Mann–Whitney U-test for continuous variables and the chi-squared test for categorical variables. Logistic regression analysis was conducted to investigate the impact of individual factors on neonatal outcome. *p*-values < 0.05 were considered significant and all tests were two-tailed. Sensitivity, specificity, and positive and negative predictive values were calculated for singleton and twin normograms. Test performance was described using receiver operating characteristic (ROC) curves, sensitivity, specificity, and predictive values. Areas under the receiver operating characteristic curve (AUC) were calculated and compared. Multilevel mixed-effect statistical models were used to evaluate growth for each twin in relation to gestational age. Separate linear models for each variable were constructed for SGA twins predicted by a singleton and a twin chart in order to allow for differences between them in both the mean and covariance structure.

## 3. Results

After excluding 152 twin pairs due to incomplete data on pregnancy or/and neonatal outcomes a preliminary analysis of the 337 twin pairs was performed and presented at the XXXIII Congress of Polish Society of Obstetricians and Gynecologists [15]. In the current analysis, a further 15 twin pairs were excluded due to uncertain gestational age. Finally, 322 twin pairs were taken into account for the analysis. The presented results are in line with previous findings. Due to strict quality control criteria, any missing piece of information or outcome measures resulted in exclusion from the study group, which was substantial (152 twin pairs). The characteristic of excluded study group was not significantly different from the main group.

Table 1 shows basic maternal characteristics of the study group. Dichorionic diamniotic (DCDA) pregnancies constituted 76.7% of our study group. The singleton chart classified as SGA 21.8% and 33.5% of dichorionic and monochorionic pregnancies, respectively, whereas the twin curve classified as SGA 21.3% and 12.2% of dichorionic and monochorionic pregnancies, respectively.

Neonatal outcome is presented in Table 2. The data included in the addendum present the distribution of the neonatal sample size in regard to a curve used and neonatal complications described. Implementation of singleton charts was associated with an increase in SGA diagnosis comparing to the twin chart—20% versus 10%, regardless of chorionicity. The group of AGA consisted of 75% and 80%, whereas LGA consisted of 5 and 10%, according to the singleton and twin chart, respectively. Our study sample corresponded with group size distribution provided by a twin normogram. Having been classified as an SGA fetus by a customized twin chart brought a substantially higher risk of developing an adverse outcome compared to the use of a singleton normogram. In our primary cohort, the number of intrauterine deaths was negligible, although in a group of liveborn cases with neonatal demise (described as a death before discharge from the hospital) 5 in 10 neonates were classified as SGA by the singleton chart and 4 by the twin chart. Addendum data describe the distributions of neonatal demises with regard to the chart used.

Test performance measures for predicting adverse neonatal outcomes for both growth charts in SGA neonates are presented in Table 3. The ROC curves for singleton and twin charts and AUC are presented in Figure 1. Singleton charts presented higher sensitivity (32.9% vs. 21.7%), yet lower specificity (83.2% vs. 93.6%) compared to twin normogram implementation. Conversely, the twin chart was characterized by higher specificity, which could be a result of selecting growth-restricted twins at the highest risk of developing neonatal complications. The predictive value for a positive result (PV+) appeared to be in favor of the twin chart (49.2% vs. 35.9%), although the predictive value for a negative result (PV−) remained comparable (81.3% vs. 80.7%) for the singleton and twin normograms, respectively. When the gender was considered, the singleton normogram showed higher sensitivity and PV− for females, with enhanced PV+ in favor of males. The twin charts showed higher sensitivity, specificity, and positive predictive value for males, having more accurate negative predictive value for female gender.

## 4. Discussion

According to our results, customized twin growth charts had higher positive predictive value for adverse neonatal outcomes in SGA newborns than singleton charts. Singleton normograms had higher sensitivity in predicting adverse outcomes, although showed lower predictive value for the study outcomes. Enhanced sensitivity shown by the singleton curve might be secondary to labelling more neonates as SGA per se—in our study group, the difference in total number of SGA neonates in both groups is substantial. The antenatal classification of SGA or selective FGR in twin pregnancy increases the risk of iatrogenic preterm birth, and, secondarily, increases the risk of prematurity-related complications.

The trade-off between the sensitivity and specificity of screening charts requires a balance and has always been a matter of debate. While the singleton chart appears to classify more pregnancies as SGA, the additional cases identified were not at significant risk of neonatal complications. The potential risk of classifying extra cases as SGA is that it will increase the rate of iatrogenic interventions and possible preterm deliveries. As twin charts predict neonatal complications with higher accuracy, the optimal prenatal care should be based on them. According to other studies, using customized charts could reduce unnecessary interventions without an increase in the rate of stillbirth [11].

The pathophysiology of twin pregnancy with regard to uterine milieu potential is essential. Due to Blickstein’s extensive studies, it has become evident that the growth curve in multiple gestation significantly differs from singletons’ increment pace [6,16]. The intrauterine growth pattern in twin pregnancy can be divided into three different stages [6,17]. In stage I, twins and singletons maintain parallel curves and grow with increment ratio of 1:1 up to 28 to 30 weeks of gestation. Stage II, promoting in utero maturity rather than weight gain, occurs between 30 and 33 weeks, and is characterized by a decrease in pace of weight increment. This period is enhanced mostly in monochorionic diamniotic (MCDA) pregnancies, as DCDA gestations do not show such highlighted deceleration [18]. The final stage, stage III, brings restoration of the 1:1 growth ratio and remains until 40 weeks of gestation, maintaining 15–20% weight deficit compared to singletons [6,16]. Additionally, twins weighting between the 5th and 10th percentile present two decelerations of the growth pace—first, between the 28th and 31st weeks, and second, later during pregnancy, which could be related to increasing placental insufficiency when approaching term [19].

Our results are more relevant regarding MCDA twins than DCDA ones. This is in line with the abovementioned discrepancies in twin fetal growth. As adaptive changes in stage II of twin fetal growth are more often expressed in MCDA gestation, using customized twin growth charts for estimating the growth of MCDA twins is more accurate in predicting neonatal complications.

In a study by Kibel et al., the authors supported the thesis of different pathophysiology of SGA in twins and singletons. SGA twins were more likely to present marginal or velamentous cord insertion rather than small placentas, hypercoiled cords, or maternal/fetal malperfusion pathology, which was encountered more frequently in SGA singleton pregnancies. The authors provided support for hypothesis that the diagnosis of SGA in twin pregnancies might be of less malignancy than in SGA singletons, reflecting the element of uterine adaptation [20]. Therefore, an implementation of customized charts for twins appears to be even more justified.

The matter of choice of appropriate charts has a substantial impact on the management of twin growth—SGA rate fluctuates between 18 and 46% if a curve for singleton is used and between 13 and 17% in the case of implementing twin-customized charts [18]. After showing a reduction in weight gain trajectory from 30 weeks’ onward, a third of near-term twins are classified as SGA using singleton charts [21].

A supremacy of twin-adjusted charts was found in a comparison of singletons’ and twins’ growth trajectories presented by Gielen et al. [22]. In their study, the birthweights of twins started to differentiate from singletons from 33 onward, with an estimated difference of 900 g at 42 weeks. After 31 weeks, the mortality rate increased as the percentiles decreased, being more pronounced below the 3rd percentile. With the usage of populational singleton charts, the authors could not draw such a clear conclusion between 32 and 37 weeks. Moreover, twins that died neonatally reached 10th percentile for the customized chart a week before the singleton chart, showing a preference for the first one [22].

In a paper by Kalafat et al., three kinds of charts—chorionicity-specific, a customized and non-customized chart for singletons—were compared. The primary outcomes were the liveborn and stillborn SGA cases detected as SGA by fetal weight estimation. In a cohort in which all three charts were compared, the rates of liveborn fetuses identified as SGA were 8.5%, 12.8% and 7.1% for the non-customized singleton, customized singleton, and twin chart, respectively. The three charts identified a similar proportion of SGA stillbirth cases regardless of the cut-off value and chorionicity; however, the twin-specific chart diminished the number of SGA fetuses, suggesting a possible reduction in unnecessary medical interventions [23]. Similarly, Mendez-Figueroa et al. presented a study in which more twins were categorized as SGA when the singleton nomogram was used (33%) compared with the twin nomogram (4%) [17]. The use of singleton normogram, showed similar neonatal mortality rate in SGA in AGA groups, contrarily to the twin curve, which revealed higher mortality rates in SGA twins.

In a study by Cordiez et al., the authors formulated a conclusion that customized charts adjusted for fetal sex and customized curves did not improve screening for SGA infants below 10th percentile [24]. Four growth curves were compared: Hadlock’s curve [13], the customized Ego’s curve [25], the EPOPé (Epidemiologie Perinatale, Obstetricale et Pediatrique) unadjusted (M0) and adjusted on the fetal sex (M1) curves [26]. Table 4 describes characteristics of the charts used in our study and the curves used by Cordiez in detection of SGA fetuses. Corresponding to our findings, the populational chart increased sensitivity, mostly due to a higher incidence of SGA. The twin chart described in our study has by far the best sensitivity amongst the mentioned curves, although PPV is lower. It is worth mentioning that in Cordiez’s study, the weight was estimated ultrasonographically in a time span less than a month before delivery, using an incoherent protocol for fetal measurement provided by sonographers of different levels of experience. In our study, an actual postnatal weight was used, and therefore a true comparison is difficult to make.

Inflating the number of SGA neonates can have a gross impact on financial aspects in health system management and might increase the number of iatrogenic preterm deliveries. On the contrary, minimizing the population at increased risk of complications carries a threat of inappropriate neonatal care and brings ethical issues. It is extremely important to distinguish fetuses that would benefit from preterm delivery and those at low risk of stillbirth, to avoid unnecessary prematurity. In our study, twin-specified nomograms decreased the rate of SGA twins, which remains consistent with similar research [17,27].

Intensive surveillance in twin pregnancy is a part of a routine prenatal care due to its nature. However, as around a third of twins after 30 weeks are diagnosed as SGA using singleton charts, the idea of customization is to acknowledge the different pattern of growth in twin charts and, therefore, reduce the number of SGA twins and avoid unnecessary interventions [23]. Ultrasound biometry shows a statistically significant reduction in twin fetal growth when compared to singletons, and this is particularly marked in the third trimester [18].

The strengths of our study are a large study sample and unified classification and management of neonatal complications, performed in a single center. To the best of our knowledge, this is the first study analyzing neonatal morbidity and mortality depending on the growth chart used. The presented unique analysis of multiple neonatal complications in SGA twins according to the singleton and twin normograms is of crucial importance for choosing which chart to use in clinical practice. Another advantage is the utilization of corresponding charts for twins and singletons, formulated by the same authors.

The main limitation of our study is its retrospective character, having an impact on outcome assessment and reporting. Our university hospital constitutes a tertiary center, and therefore the study group mostly consisted of referred cases from the maternal high-risk population, which possibly influenced the fetal weight and the reported risk of neonatal outcome. A substantial ratio of iatrogenic preterm deliveries obscuring the natural history of many pregnancies is considered as an additional drawback. Moreover, given the retrospective nature of the study, data regarding confounding factors (for example, steroid administration) or classifying the severity of fetal growth restriction were unavailable or incomplete. Although we excluded pregnancies complicated by TTTS or TAPS, underlying vascular pathology in monochorionic pregnancies might have inflated the total number of SGA twins.

Due to the lack of results from randomized studies, the controversy around singleton and twin growth charts for twin growth has not been resolved. Implementation of singleton normograms undoubtedly leads to an increase in the diagnosis of number of SGA twins. The published research on the effectiveness of different growth charts in predicting neonatal outcome is conflicting [18,28,29].

## 5. Conclusions

The estimation of twin growth with the customized charts provides better prognosis of adverse neonatal outcomes in SGA group comparing to the singleton nomograms, and this was more pronounced for MCDA than DCDA twins; however, large multi-center prospective studies are needed to show a virtual accuracy of different normograms.

## Figures and Tables

**Figure 1 ijerph-18-02016-f001:**
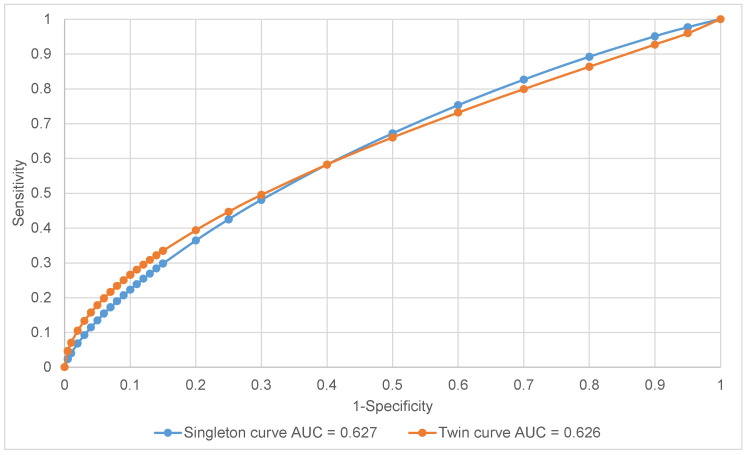
Receiver operating characteristic (ROC) curves—total.

**Table 1 ijerph-18-02016-t001:** Maternal characteristics of the study group.

	Included		Excluded
	Total *N* = 322	Total (%)	Total *N* = 184
Ethnicity			
Caucasian	312	96.9	182
Other	10	3.1	2
Maternal age (years)			
<20	1	0.3	2
20–35	259	80.4	137
>35	62	19.3	34
Primiparity	221	68.6	83
Parity ^Ϯ^	1.39	0.72	1.39
Chorionicity			
Dichorionic diamniotic	247	76.7	12
Monochorionic diamniotic	75	23.3	19
Smoking during pregnancy	17	5.3	
ART	109	33.9	
Diabetes mellitus (pre- and gestational)	58	18.0	
Hypertensive disease of pregnancy	99	30.7	
Intrahepatic cholestasis of pregnancy	12	3.7	
Gestational age at delivery (weeks)			
24.0–27.6	2	0.6	
28.0–31.6	24	7.5	
32.0–36.6	187	58.1	
≥37.0	109	33.9	
Caesarean delivery	210	65.2	
Vaginal delivery	112	34.8	

^Ϯ^—mean/ + SD; ART—assisted reproduction techniques.

**Table 2 ijerph-18-02016-t002:** Neonatal outcomes of the study group.

	Singleton Normogram					Twin Normogram				
	SGA (*N* = 131)	AGA (*N* = 481)	LGA (*N* = 32)	RR SGA vs. AGA	RR (95% CI)	SGA (*N* = 63)	AGA (*N* = 518)	LGA (*N* = 63)	RR SGA vs. AGA	RR (95% CI)
Composite neonatal morbidity	47	94	4			31	102	11		
Apgar score < 8 at 5 min	14	12	1			9	16	2		
CPAP	23	63	3	1.9	1.2–2.9	16	66	7	2.5	1.5–4.3
Mechanical ventilation	19	28	1	4.3	1.9–9.5	12	33	3	4.6	2.0–11.0
NICU admission	45	94	4	2.5	1.4–4.6	31	102	10	3.1	1.6–6.3
IVH (grade 3 or 4)	1	3	0	1.3	0.8–2.3	1	3	0	2.0	1.1–3.7
NEC	2	0	0	2.5	1.3–4.6	2	0	0	3.0	1.5–6.1
Sepsis	3	2	0	2.0	1.2–3.4	3	2	0	2.5	1.3–4.6
Pneumonia/inborn infection	28	50	2	1.2	0.1–11.9	19	56	5	2.7	0.3–26.8
Neonatal death	5	5	0			4	4	2		

RR—risk ratio; CI—confidence interval; SGA—small for gestational age; AGA—appropriate for gestational age; LGA—large for gestational age; CPAP—continuous positive airway pressure; NICU—Neonatal Intensive Care Unit; IVH—intraventricular hemorrhage; NEC—necrotizing enterocolitis.

**Table 3 ijerph-18-02016-t003:** Sensitivity, specificity, positive and negative predictive values for singleton and twin normograms in detection of SGA fetuses.

Singleton Normogram	Total	Females	Males	MCDA	DCDA
Sensitivity	32.9%	35.5%	28.9%	48.9%	25.5%
Specificity	83.2%	82.3%	87.4%	77.1%	84.8%
Positive predictive value	35.9%	34.9%	44.0%	47.8%	29.4%
Negative predictive value	81.3%	82.6%	78.3%	77.9%	82.2%
*p*-value (for this type of test)	0.17	0.18	0.13	0.23	0.15
**Twin Normogram**					
Sensitivity	21.7%	17.7%	22.1%	33.3%	16.3%
Specificity	93.6%	94.4%	96.3%	88.6%	94.9%
Positive predictive value	49.2%	45.8%	68.0%	55.6%	44.4%
Negative predictive value	80.7%	81.0%	77.6%	75.6%	82.1%
*p*-value (for this type of test)	0.06	0.06	0.04	0.11	0.05

SGA—small for gestational age; MCDA—monochorionic diamniotic; DCDA—dichorionic diamniotic.

**Table 4 ijerph-18-02016-t004:** Comparison of sensitivity, specificity, and positive and negative predictive values of selected charts in detection of SGA fetuses.

	Singleton Normogram	Twin Normogram	Hadlock [13]	Customised Ego [25]	EPOPé MO [26]	EPOPé M1 [26]
Sensitivity	32.9%	21.7%	67.3%	63.0%	59.9%	57.4%
Specificity	83.2%	93.6%	80.0%	82.3%	83.5%	83.2%
PPV	35.9%	42.9%	63.7%	65.0%	65.5%	64.1%
NPV	81.3%	80.7%	82.4%	80.9%	79.9%	78.9%

PPV—positive predictive value; NPV—negative predictive value; EPOPé—Epidemiologie Perinatale, Obstetricale et Pediatrique [25].

## Data Availability

The data presented in this study are available on request from the corresponding author.

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
