# Peer review of "Predictive Accuracy of Singleton Versus Customized Twin Growth Chart for Adverse Perinatal Outcome: A Cohort Study"

_ijerph, 2021, doi:10.3390/ijerph18042016_

Round 1

Reviewer 1 Report

Urszula Sarzyńsk et al

Predictive accuracy of singleton versus customized twin growth chart for adverse perinatal outcome: a cohort study. 

Author Response

Honourable Reviewer,

Thank you for your time to read and analyse our manuscript and for a positive opinion.

Reviewer 2 Report

The paper is clearly presented with specific and relevant objectives for twin care. I have minor comments to be addressed 

Introduction 

  • Authors state that the slow in fetal growth in third trimester is an adaptative situationWhat evidence does support this informationWheather if it is adaptative or pathological is still controversial 
  • Recent MCDA charts have been publised and not referred in the text  Gender-Specific Antenatal Growth Reference Charts in Monochorionic Twins.Torres X, Bennasar M, Eixarch E, Rueda C, Goncé A, Muñoz M, Marimón E, Martínez JM, Gratacós E, Figueras F.Fetal Diagn Ther. 2018;44(3):202-209. doi: 10.1159/000484555. Epub 2017 Dec 21.PMID: 29268248 

Methods: 

  • Authors excluded Pregnancies delivered <24 weeks gestationWhat is the reasen? 
  • Why did they exclude fetal demiseIt is considered an adverse neonatal outcome, so those cases should be included in order not to make a bias 

Results: 

  • Was there any difference between the excluded population and the included populationThis should be evaluated in order to avoid bias 
  • Implementation of singleton charts was associated with an in- 126 crease in SGA diagnosis comparing to twin chart - 20% versus 10%”: Which is the difference between line 120 and line 126? 
  • By their results the detection of a bad outcome (neonatal deathof using twin charts seems lower than using singleton..... 5/10 vs 4/10. There is a need to evaluate what is needed... is it better to increase sensitivity or specificityIf we refer to neonatal death sensitivity is required . However if we are talking on preterm birth specificity should be preferred, in my opinionThis should be discussed. 

The results are more relevant regarding MC than DC twinsThis should be discussed in results and conclusions 

Author Response

Honourable Reviewer,

Thank you for your time and all the valuable suggestions which helped us improve the paper. Here are the answers to your comments:

  1. „Authors state that the slow in fetal growth in third trimester is an adaptative situation. What evidence does support this information? Wheather if it is adaptative or pathological is still controversial”

Indeed, whether it is a pathological or adaptative situation still remains controversial, nevertheless studies show a different growth pattern of twins and singletons and, in our opinion, it should be considered in clinical management. It has been highlighted in the manuscript. Growth discrepancies depending on chorionicity and fetal gender has been mentioned as well.

  1. „Recent MCDA charts have been publised and not referred in the text  Gender-Specific Antenatal Growth Reference Charts in Monochorionic Twins.Torres X, Bennasar M, Eixarch E, Rueda C, Goncé A, Muñoz M, Marimón E, Martínez JM, Gratacós E, Figueras F.Fetal Diagn Ther. 2018;44(3):202-209. doi: 10.1159/000484555. Epub 2017 Dec 21.PMID: 29268248”

Thank you very much for this comment. The references have been updated with the study.

  1. „Authors excluded Pregnancies delivered <24 weeks gestation. What is the rease? Why did they exclude fetal demise? It is considered an adverse neonatal outcome, so those cases should be included in order not to make a bias”

Prenatally, growth charts should refer to ideal circumstances, under which fetuses reach the maximum of growth potential. In all of the above-mentioned situations growth retardation may play a major pathophysiological role, therefore those cases were excluded from analysis. In a case of fetal demise in a twin gestation, the exact time of the event is difficult to establish and a birth weight of a demised fetus may be falsely decreased due to delay between the time of demise and delivery. It is rather difficult to estimate the percentage of FGR in demised fetuses in a retrospective analysis and, therefore, all the cases of in utero deaths were excluded from analysis. An arbitrary gestational age criterion – deliveries from 24 weeks onward, commonly used in reviewed articles, was set due to viability threshold in neonatal care in majority of countries worldwide. Moreover, in case of miscarriage before 24 weeks the patients were not routinely hospitalized, only if induction of labour was necessary. This comment was added to the manuscript.

  1. „Was there any difference between the excluded population and the included population? This should be evaluated in order to avoid bias”

Due to strict quality control criteria, any missing piece of information or outcome measures resulted in exclusion from the study group, which was substantial (152 twin pairs). The characteristic of excluded study group was not significantly different from the main group (see Table 4). This information was added to the manuscript.

  1. “Implementation of singleton charts was associated with an increase in SGA diagnosis comparing to twin chart - 20% versus 10%”: Which is the difference between line 120 and line 126? 

Line 120 discriminates between chorionicity, line 126 regardless of chorionicity. This explanation was added to the manuscript.

  1. „By their results the detection of a bad outcome (neonatal death) of using twin charts seems lower than using singleton..... 5/10 vs 4/10. There is a need to evaluate what is needed... is it better to increase sensitivity or specificity? If we refer to neonatal death sensitivity is required . However if we are talking on preterm birth specificity should be preferred, in my opinion. This should be discussed.”

The trade-off between sensitivity and specificity of screening charts requires a balance and has always been a matter of debate. While the singleton chart appears to classify more pregnancies as SGA, the additional cases identified were not at significant risk of neonatal complications. The potential risk of classifying extra cases as SGA is that it will increase the rate of iatrogenic interventions and possible preterm deliveries. As twin charts predict neonatal complications with higher accuracy, the optimal prenatal care should be based on them. This stays in agreement with other authors’ opinion and was added to the manuscript.

  1. The results are more relevant regarding MC than DC twins. This should be discussed in results and conclusions 

Our results are more relevant regarding MCDA twins than DCDA ones. It stays in line with the above mentioned discrepancies in twin fetal growth. As adaptive changes in stage II of twin fetal growth are more expressed in MCDA gestation, using customized twin growth charts for estimating growth of MCDA twins is more accurate in predicting neonatal complications. This paragraph was added to the discussion part.

Reviewer 3 Report

The authors aimed to assess the fetal growth in twin gestations in relation to singleton charts and customised twin charts respectively, followed by a comparison of frequency of neonatal complications in small-for-gestational-
age (SGA) twins. The comments are as follows:

Major comments

  1. The introduction does not provide sufficient information and needs to be improved.
  2. Methods: To simple. Please introduce the design of the study type (cohort?), measures, covariates, and quality control.If this is a cohort study, logistic regression analysis is not appropriate. Please calculate RR instead of OR. Linear mixed models should also be placed and introduced in the Statistical analysis. For the repeated measurement  data, does the authors consider using GEE models? 
  3. After excluding 152 twin pairs due to incomplete data on pregnancy or/and neonatal outcomes a preliminary analysis of the 337 twin pairs was performed and presented at the XXXIII Congress of Polish Society of Obstetricians and Gynecologists. There are large proportion on the excluding twin pairs. There might be selection bias. Please compare the characteristics of excluding and including twin pairs.

Minor comments:

     1.please adds ROC curves.

Author Response

Honourable Reviewer,

Thank you for your time and all the valuable suggestions which helped us improve the paper. Here are the answers to your comments:

  1. The introduction does not provide sufficient information and needs to be improved.

Thank you very much for this comment. The introduction it has been updated.

  1. Methods: To simple. Please introduce the design of the study type (cohort?), measures, covariates, and quality control.If this is a cohort study, logistic regression analysis is not appropriate. Please calculate RR instead of OR. Linear mixed models should also be placed and introduced in the Statistical analysis. For the repeated measurement  data, does the authors consider using GEE models?

Thank for all the comments, it helped us improve the manuscript. The text has been modified according to your suggestions.

  1. After excluding 152 twin pairs due to incomplete data on pregnancy or/and neonatal outcomes a preliminary analysis of the 337 twin pairs was performed and presented at the XXXIII Congress of Polish Society of Obstetricians and Gynecologists. There are large proportion on the excluding twin pairs. There might be selection bias. Please compare the characteristics of excluding and including twin pairs.

The basic characteristic was included. However majority of cases were excluded due to incomplete data, therefore an analysis of excluded cases is not possible.

  1.    please adds ROC curves.

The ROC curve has been added.